# Diagnostic and Prognostic Ability of Pancreatic Stone Protein: A Scoping Review

**DOI:** 10.3390/ijms25116046

**Published:** 2024-05-31

**Authors:** Christos Michailides, Themistoklis Paraskevas, Silvia Demiri, Charikleia Chourpiliadi, Konstantinos Papantoniou, Ioanna Aggeletopoulou, Eleni Konstantina Velissari, Maria Lagadinou, Christos Triantos, Dimitrios Velissaris

**Affiliations:** 1Department of Internal Medicine, University Hospital of Patras, 26504 Patras, Greece; christos.mich1@gmail.com (C.M.); themispara@hotmail.com (T.P.); sylviantemiri@gmail.com (S.D.); hara.hourpiliadi@gmail.com (C.C.); g.papanton@yahoo.gr (K.P.); elvelissari@yahoo.com (E.K.V.); m_lagad2004@yahoo.gr (M.L.); dimitrisvelissaris@yahoo.com (D.V.); 2Division of Gastroenterology, Department of Internal Medicine, University Hospital of Patras, 26504 Patras, Greece; iaggel@upatras.gr; 3Medical School of Patras, University of Patras, 26504 Patras, Greece

**Keywords:** pancreatic stone protein, diagnosis, prognosis, infectious diseases, bacterial infection, sepsis, biomarker

## Abstract

Pancreatic stone protein (PSP) is an acute-phase reactant mainly produced in response to stress. Its diagnostic and prognostic accuracy for several types of infection has been studied in several clinical settings. The aim of the current review was to assess all studies examining a possible connection of pancreatic stone protein levels with the severity and possible complications of patients diagnosed with infection. We performed a systematic search in PubMed, Scopus, the Cochrane Library and Clinicaltrials.gov to identify original clinical studies assessing the role of pancreatic stone protein in the diagnosis and prognosis of infectious diseases. We identified 22 eligible studies. Ten of them provided diagnostic aspects, ten studies provided prognostic aspects, and another two studies provided both diagnostic and prognostic information. The majority of the studies were performed in an intensive care unit (ICU) setting, five studies were on patients who visited the emergency department (ED), and three studies were on burn-injury patients. According to the literature, pancreatic stone protein has been utilized in patients with different sites of infection, including pneumonia, soft tissue infections, intra-abdominal infections, urinary tract infections, and sepsis. In conclusion, PSP appears to be a useful point-of-care biomarker for the ED and ICU due to its ability to recognize bacterial infections and sepsis early. Further studies are required to examine PSP’s kinetics and utility in specific populations and conditions.

## 1. Introduction

The efficacy of biomarkers in clinical medicine has risen during implementation over the years, and the contribution of biomarkers to the early diagnosis and effective treatment of many diseases is being progressively more intensively investigated [1]. Bacterial infections and sepsis are two of the leading causes of prolonged hospitalization, mortality, and undesired outcomes in the clinical course of patients worldwide. Several biomarkers such as C-reactive protein (CRP), interleukin 6 (IL-6), and procalcitonin (PCT) are widely used in clinical practice for the early diagnosis and prediction of clinical outcomes in patients with infection [2]. Pancreatic stone protein (PSP) is an acute-phase reactant, which is mainly produced in the exocrine pancreas as well as in other organs in response to systematic stress, with its levels increasing proportionally to the severity of inflammation. PSP triggers polymorphonuclear cell activation by binding to their surface, thus contributing to a patient’s inflammatory response [3]. Since its introduction as a diagnostic and prognostic tool for sepsis and systemic inflammatory response syndrome (SIRS) in 2004, several studies have examined its diagnostic and prognostic accuracy in several clinical settings. The aim of this study was to systematically review all studies examining a possible connection of PSP levels to the severity and possible complications during and after hospitalization of patients diagnosed with infection.

## 2. Methods

### Eligibility Criteria, Searches, and Data Collection

In this scoping review, we included studies that measured pancreatic stone protein in patients hospitalized for infectious causes and determined its diagnostic or prognostic ability. We did not exclude studies based on type of infection or hospital setting. Randomized-controlled trials, cohort studies, case-control and cross-sectional studies were eligible for inclusion. We excluded reviews, case series, case reports, expert opinions, and animal studies.

We searched PubMed, Scopus, the Cochrane Library, and Clinicaltrials.gov between August and November 2023 using the following terms: (“pancreatic stone protein” or “lithostathin” or “regenerating protein”). After deduplication, a two-step screening process was conducted independently by two authors (CC and SD), and conflicts were resolved by a third author (TP). Relevant data were extracted from each included study in a pre-defined excel format (title, publication year, first author, study location, study design, setting, prognostic/diagnostic evaluation, site of infection). Additional data for each study (PICOTS) and numerical data were also tabulated and summarized graphically.

## 3. Results

### 3.1. Flow Diagram

After deduplication, we examined 649 studies. We included 24 studies from the PubMed and Embase databases, one study from clinical trials, and one from the Cochrane Library, but some studies were duplicates and so were removed. Afterward, we did a full-text assessment and Ventura et al. [4] because it was a case report, and we excluded the study by Keel 2009 et al. [3] due to its lack of clinical interest. Consecutively, we concluded with 22 studies (Figure 1).

### 3.2. Type of Studies

In this scoping review, we included ten diagnostic studies and two studies that were both diagnostic and prognostic (Table 1).

The studies of de Hond et al. and Garcia de Guadiana-Romualdo 2017 et al. both examine the diagnostic value of PSP for detecting sepsis in patients with suspicion of infection in the emergency department [17,18]. The study by Garcia de Guadiana-Romualdo 2018 et al. is also conducted in an ED setting [19]. However, it focuses on the value of PSP in the diagnosis of infection in cancer patients with febrile neutropenia.

The study of de Hond et al. investigates the diagnostic ability of PSP in an ED setting. After the exclusion of COVID-19 patients, PSP can discriminate patients with sepsis from patients with uncomplicated infection (*p* = 0.032) and no infectious group (*p* = 0.022), respectively, although it does not manage to differentiate sepsis from uncomplicated infection when COVID-19 patients were included (*p* = 0.43). For this outcome, PSP performs better than CRP and white blood cells. Furthermore, PSP concentrations are significantly higher in septic patients compared to patients with no infection or uncomplicated infection in both cohorts [17].

The authors also perform an analysis merging the no infection and uncomplicated infection patient groups to closely examine the discriminative ability of PSP for sepsis. Through this, no remarkable outcomes are identified for the whole cohort whereas a moderate AUC of 0.65 is yielded in the cohort excluding COVID-19 patients. In the multivariate analysis, including several predictors of clinical relevance, only three variables remain clinically significant: age, COVID-19 infection, and PSP. The AUC of this model is 0.69, with a negative predictive value of 84.4% and a positive predictive value of 100%. Lastly, the authors conduct a separate analysis incorporating all the false-negative and false-positive patients to determine which of the patient characteristics alter PSP’s diagnostic accuracy. The results reveal that PSP demonstrates inadequate performance in immunocompromised, oncological, and hematological patients. Overall, the authors comment that PSP is a more suitable marker of general inflammation rather than of sepsis in particular [17].

Garcia de Guadiana-Romualdo 2017 et al. assess the diagnostic value of PSP in the ED. In their results, PSP displays a significant accuracy in differentiating infected patients from non-infected patients, patients with uncomplicated infection, patients with and sepsis, and it can also identify patients with bacteremia. Of note, PSP concentrations are significantly elevated in patients classified in the sepsis group [18].

Garcia de Guadiana-Romualdo 2018 et al. examine PSP as a diagnostic marker of infection in a cohort of cancer patients with chemotherapy-associated febrile neutropenia (FN). The PSP values are significantly higher among patients with infection compared to those without. No significant differences are observed between clinically documented infection (CDI) groups and microbiologically documented infection (MDI) groups. Similarly, there is no statistically significant difference between bacteremic and non-bacteremic episodes of the MDI group. In the ROC curve analysis, PSP features a good diagnostic accuracy for infection, with an AUC of 0.751, but does not outperform PCT [19].

The studies of Niggerman et al., Klein 2020 et al., and Klein 2021 et al. include a special population of severely burned patients [20,21,22]. Specifically, Klein 2020 et al. examine the levels of PSP and other biomarkers in patients with inhalation injury [22].

Niggerman et al. focus on investigating, among others, the diagnostic abilities of PSP in identifying sepsis in burned patients in relation to the incidence and time-related occurrence of septic events according to three separate sepsis definitions. Irrespective of the definition, PSP is the only biomarker to demonstrate a highly significant interaction between time and group (septic versus non-septic), with a *p* value of <0.001. Furthermore, PSP concentrations show a significant rise in septic patients, occurring sharply 48 h after admission up to 72 h before sepsis diagnosis, with a 3.3–5-fold increase across all three definitions. According to the sepsis 3 definition, infected and non-infected patients have similar PSP levels, but PSP cannot discriminate between infected patients with and without sepsis [20].

The study of Klein 2020 et al. aims to elucidate the influence of inhalation injury and inhalation-injury-induced acute respiratory distress syndrome (ARDS) on PSP kinetics and to interpret its effect on sepsis diagnosis. The authors conduct an observational study in which they include 90 burn-trauma patients and retrospectively examine their clinical parameters collected over the course of 14 consecutive days [22].

Regarding the effect of inhalation injury on PSP kinetics, the authors find no difference in PSP levels between the inhalation injury and the non-inhalation injury group at baseline. They also note that there is no association between the levels and the grades of inhalation injury. However, PSP is the only biomarker to exhibit a significant time and inhalation-injury status correlation (*p* < 0.001), as its levels display a steep increase over time. The authors state that this statistically significant finding is possibly due to the susceptibility of inhalation injury patients to respiratory tract infections. Even though the incidence rates of septic pneumonia are higher in the inhalation-injury group compared to the non-inhalation-injury patients, the authors do not prove a significant correlation (*p* = 0.099) [22].

Klein 2020 et al. mention that ARDS occurs in approximately 40% of inhalation injury patients in ICU. They refer to “direct ARDS”, which is directly induced by inhalation injury and presents in a matter of 7 days as suggested by the Berlin definition. In their study, they conclude that the occurrence of ARDS is not correlated with the severity of inhalation injury during the first week of admission. As aforementioned, PSP levels display an increase over time, but it cannot be associated with ARDS as it may occur during the first week after trauma; therefore, it seems to be due to sepsis progression. The authors disclose that the ARDS group was too small (no. = 8) for a statistically significant conclusion to be made [22].

Both the publications by Tschuor et al. and Fischer et al. are study protocols. Tschuor et al. provide a protocol for a multi-center cohort study. The authors aim to determine the diagnostic value of PSP regarding acute appendicitis, due to the lack of diagnostic tests that confirm the appendiceal inflammation. Fischer et al. design a protocol for a prospective, monocentric cohort study that will preoperatively recruit patients undergoing major abdominal surgery. The purpose of this study is to evaluate the roles of PSP and PAP (pancreatitis-associated protein) in differentiating between postoperative septic complications and simple inflammatory responses [16,23].

Pugin et al. 2021, Klein et al. 2015, and Llewelyn 2013 et al. examine the diagnostic accuracy of PSP in septic patients admitted to the ICU [24,25,26]. Pugin et al. conduct a multinational, prospective, blinded observational clinical study to investigate the effectiveness of serial measurements of CRP, PCT, and PSP in early sepsis detection among patients in the ICU. This study highlights the notable association between increasing PSP levels in the three days preceding a clinical sepsis diagnosis, highlighting the potential of PSP and other host protein biomarkers for early sepsis identification in ICU patients [24].

In the study of Klein 2015 et al., only PSP levels demonstrate the ability to effectively distinguish between infection and the surgical-trauma-associated inflammation of patients in the postoperative phase following cardiac surgery. When evaluating different types of cardiac surgical procedures, authors do not detect any notable differentiation in PSP levels at postoperative days 1–3. Regarding baseline characteristics, patients with type II diabetes mellitus exhibit significantly higher PSP levels compared to non-diabetic individuals; no correlation is observed between PSP levels and obesity (BMI > 30). However, PSP levels are notably elevated in elderly patients above the median age of 67 years. The difference in PSP levels among diabetic patients loses statistical significance after adjusting for age. Furthermore, PSP levels are significantly higher in infected and septic patients during the first 72 h following cardiac surgery compared to those with uncomplicated recoveries. However, there are no significant differences in PSP, CRP, or WBC levels during postoperative days 1–3. Analyzing the impact of surgical trauma in relation to biomarker dynamics, patients undergoing sternotomy present with a significantly steeper increase in postoperative PSP levels compared to those subjected to a minimally invasive approach. The type of surgical technique does not have a significant impact on the postoperative distribution of CRP and WBC levels. Receiver operating characteristic (ROC) curve analysis is employed for PSP on postoperative days 1–3. Notably, PSP at postoperative day 2 exhibits the highest sensitivity and specificity, with an AUC of 0.765 (95% CI 0.621–0.877). In contrast, traditional inflammatory markers show limited predictive capability for infection [25].

Llewelyn 2013 et al. evaluate the efficacy of several biomarkers to recognize sepsis or severe sepsis by assessing the area under the ROC curve. Regarding PSP, there is a clear correlation between plasma levels and the severity of sepsis, but not the severity of non-infectious SIRS in patients admitted to the ICU. This study also reveals that PSP, sCD25, and PCT are all efficient in sepsis prediction. PSP and PCT also exhibit the ability to reflect the severity of sepsis [26].

In this scoping review, we also included ten prognostic studies and two studies that were both diagnostic and prognostic (Table 2).

In the study of Ping Hu et al., the authors assess the prognostic value of PSP/Reg in regard to sepsis progression. They specifically focus on sepsis-induced multiple organ dysfunction syndrome (MODS) and mortality rates. Regarding sepsis severity, the PSP/Reg values of septic shock patients are found to be higher than those of severe sepsis patients and also to be correlated with the initial sequential organ failure assessment (SOFA) score. Notably, when the SOFA score exceeds 5 points, patients progress to MODS. It is reported that patients with MODS exhibit higher circulating values of PSP/Reg compared to those without MODS (*p* = 0.001). Therefore PSP/Reg levels display a correlation with disease severity. Additional evidence supports this statement, such as the Kaplan- Meier analysis of elevating biomarker levels that demonstrates a strong correlation with the 28-day mortality rates (*p* < 0.001).

The authors also examine the relationship between PSP/Reg levels and the need for organ support such as vasopressors on admission, long-term vasopressor administration, mechanical ventilation, or renal replacement. Through this examination, a significant correlation is identified. Ping Hu et al. conclude that PSP/Reg is an independent risk factor for progression to MODS 48–72 h after admission and for 28-day mortality (*p* < 0.001), and they confirm its predictive value through ROC analysis and further sub-analysis of severe sepsis and septic shock patients [5].

Both Van Singer et al. and Lagadinou et al. aim to evaluate the prognostic value of PSP in COVID-19 patient cohorts [7,8]. The study of Lagadinou et al. concentrates solely on PSP, whereas the study of Van Singer et al. extends its analysis by incorporating CRP values and assessing the accuracy of the qSOFA and CRB-65 scores of bedside clinical severity [7,8].

Lagadinou et al. investigate the prognostic value of PSP with regard to 28-day in-hospital mortality, length of hospital stay, and utilization of non-invasive mechanical ventilation. In this study, PSP yields an AUC value of 0.588 and 0.545 for 28-day in-hospital mortality and the need for non-mechanical ventilation, respectively. As evidenced by these data, PSP is not a suitable biomarker for the prediction of such outcomes but it can effectively serve in the identification of patients at high risk for prolonged hospitalization, as demonstrated by the AUC value of 0.800 [7].

The study of Van Singer et al. also focuses on in-hospital mortality but restricts it to a 7-day timeframe since ED admission. Their findings indicate an excellent prognostic accuracy of PSP (AUC = 0.833), especially when combined with the CRB-65 clinical severity score (AUC = 0.95). They also evaluate the performance of PSP in relation to ICU admission. The results show that CRP outperforms PSP in predicting 7-day ICU admission, as it demonstrates an AUC value of 0.74 compared to PSP’s value of 0.51 [8].

In the study of Garcia de Guadiana-Romualdo 2019 et al. the authors aim to evaluate the predictive value of PSP for 28-day mortality in critically ill patients. PSP levels on day 2 display a statistically significant increase in the non-survival group compared to the survival group. The findings of the 28-day prediction model show that the combination of PSP and lactate has an AUC–ROC of 0.796. The performance of the SOFA score alone has an AUC–ROC of 0.826, but in combination with both biomarkers, achieves an AUC–ROC of 0.866. They also note that the difference between the two values approaches statistical significance (*p* = 0.080). On the other hand, the results of the reclassification analysis reveal that net reclassification improvement (NRI) favors the combination of PSP and lactate over the SOFA score in identifying non-survivors (NRI: 2.7; *p* = 0.008). The Kaplan–Meier analysis similarly indicates that mortality rates are higher in patients with increased levels of PSP and lactate. Additionally, a Cox regression analysis, accounting for several factors, reveals that baseline PSP is the only independent predictor for 28-day mortality. In summary, PSP has a significant discriminative value in predicting 28-day mortality in critically ill patients [9].

Yok-Ai Que 2015 et al. focus on developing and validating a predictive model of in-hospital mortality from an independent derivation and validation cohort of critically ill patients with sepsis or septic shock. It is revealed that PSP levels are significantly elevated in patients with septic shock compared to those with sepsis in both cohorts, in addition to being significantly lower in surviving patients. PSP demonstrates a moderate accuracy in predicting death only in the derivation cohort, with an AUC of 0.665. The authors develop a model that combines PSP and PCT with SAPS II to give an AUC of 0.710, and a slightly superior combination of PSP and PCT with APACHE II yields an AUC of 0.721. However, when validating the outcome of the predictive models, a reduced performance is observed with AUCs of 0.629 and 0.637, respectively. Yok-Ai Que 2015 et al. highlight the importance of PSP in these predictive models with an illustration of the predicted probability of death as a function of PSP [10].

Gukasjan R et al. aim to assess the effectiveness of PSP in predicting sepsis-related postoperative complications and death within ICU cohorts. The authors recruit postoperative patients with a proven diagnosis of secondary peritonitis. Their findings from the univariate analysis suggest that PSP is the only biomarker with a significant correlation to the presence of organ failure (single, multiple) and mortality in the ICU, with AUCs of 0.81, 0.75, and 0.78, respectively. Amongst the blood parameters and clinical scores, PSP and SOFA outperform all other parameters to predict ICU death. The prognostic value of PSP is further established in a multivariate stepwise regression analysis, in which PSP is proven to be the only independent predicting factor for death in the ICU (*p* = 0.013). PSP has the best correlation compared to all other biomarkers with regard to severity scores, and it is the best at predicting the severity of the disease and renal failure [11].

In the study of Scherr A et al., the authors investigate the correlation between PSP/reg and acute exacerbation of chronic obstructive pulmonary disease (COPD), focusing on the prediction of positive bacterial infections. According to their findings, PSP/reg levels are elevated in patients with acute exacerbations of COPD (AECOPD) compared to stable COPD patients and healthy control subjects (*p* < 0.01). However, the biomarker has a weak correlation with the Charlson age and condition-related score (r = 0.30, *p* < 0.01) and exhibits a subtle level of differentiation between separate Anthonisen risk stratification categories. Interestingly, PSP/reg levels remain mostly stable in the whole study population throughout hospitalization, regardless of antibiotic or steroid treatment. Nonetheless, antibiotic treatment has an impact only on patients with culture-positive AECOPD, as implied by the decreasing PSP/reg values [12].

Through Scherr A et al.’s findings, it is revealed that PSP/reg is a weak predictor for the length of hospital stay that has no association with the need for ICU admission (*p* = 0.24), in-hospital mortality (*p* = 0.90), re-exacerbation (*p* = 0.45), or re-hospitalization after discharge (*p* = 0.65). Of note, a retrospective correlation is observed between the development of pneumonia over the course of AECOPD and elevated PSP/reg levels on admission. Furthermore, PSP/reg levels, among other parameters, are associated with 2-year mortality on logistic regression analysis. The authors also perform a Kaplan–Meier analysis investigating the cumulative risk of death. The results indicate that high PSP/reg levels are related to an increase in 2-year mortality rates.

Regarding the predictive value of PSP/reg in patients with AECOPD and positive bacterial cultures, PSP/reg effectively identifies patients with positive sputum cultures as evidenced by the elevated levels of the biomarker compared to the similar lower levels of the negative-culture AECOPD and stable COPD patients. A multivariate analysis further supports those data, as PSP/reg is proven to be the only independent predictor of positive sputum microbiology in AECOPD patients. Notably, when combined with FEV1% it remains an independent predictor irrespective of the subgroups of patients who have received antibiotics upon hospital admission. The performance of PSP/reg as a biomarker is especially enhanced when used with the marker of the presence of discolored sputum, as it then reaches a sensitivity rate of 97% [12].

Yok-Ai et al. and Que 2012 et al. investigate the prognostic accuracy of PSP/reg in identifying the mortality risk of patients with sepsis or septic shock. The median concentrations of PSP/reg in patients with septic shock (343.5 ng/L) are five-fold higher than those of severe sepsis patients (73.5 ng/L) among those admitted to the ICU within 24 h following hospital admission. Remarkably, the authors found that PSP/reg values exhibit a significant difference between survivors and non-survivors irrespective of disease severity (*p* = 0.02). The PSP and SAPS scores are the only parameters that differentiate between survivors and non-survivors. Their findings also prove that day-1 PSP is the only parameter associated with mortality in patients with septic shock (*p* = 0.049). Furthermore, PSP/reg demonstrates a more linear and uniform distribution compared to other biomarkers in the overall range of in-hospital mortality probability estimation. Regarding age adjustment, the probability of in-hospital mortality increases by 0.16 (SD of 0.07) per 100 ng/mL change in PSP/reg values (*p* = 0.03). When examining only the septic shock patient cohort, the coefficients are 0.16 (SD of 0.08) per 100 ng/mL increase in PSP/reg values (*p* = 0.048). Lastly, another significant correlation is revealed between the age-adjusted ORs of mortality among patients with septic shock and PSP/reg quartiles. The former increase continuously across the increase of the latter (*p* = 0.02) [13].

The study of Boeck L et al. examines a specific patient cohort that is suffering from ventilator-acquired pneumonia (VAP). This study aims to assess the prognostic ability of PSP/reg in VAP with regard to the survival and mortality rates. The PSP/reg values on day-0 are weakly correlated with age and VAP and not associated with sex, co-morbidities, or gas exchange. However, on day a significant correlation occurs between the PSP/reg and SOFA scores on day-0 (*p* < 0.001) for both survivors and non-survivors. Specifically, the PSP/reg values of day-0 are significantly elevated in non-survivors, indicating a strong relation with mortality risk (*p* = 0.011). During the following 7 days of VAP onset, PSP/reg levels are moderately linked to daily SOFA scores and are not correlated to the duration of antibiotic therapy. Furthermore, the decreasing rate of PSP/reg values has no association with day and group (survivors and non-survivors).

The results of the univariate logistic regression highlight the PSP/reg values of day-0 and day-7 as the most suitable predictors of survival, as they demonstrate ORs and *p* values of 1.60, *p* = 0.022, and 2.36, *p* = 0.007, respectively. The most accurate threshold to predict survival is estimated to be 24 ng/mL, which gives a sensitivity of 36% and a specificity of 100%. In contrast, the most significant parameter concerning the prediction of death is the PSP/reg cut-off value of 177 ng/mL at day-7 after VAP onset, with 58% and 91% sensitivity and specificity, respectively. Lastly, the ROC analysis regarding VAP from onset to day-7 provides the AUCs of 0.69 and 0.76, for survival and mortality, respectively [14].

In a prospective observational study by Michailides et al. PSP has been examined for its prognostic ability for sepsis, hospital readmission, need of treatment escalation, need of surgery treatment, mortality and length of hospital stay (LOS) among patients with intrabdominal infection (IAI), using a measurement within the first 24 h since admission. PSP fails to predict mortality, LOS, or need of surgery treatment among those patients; however it successfully predicts sepsis, readmission, and treatment escalation, with AUCs of 0.694, 0.899, 0.862, respectively. PSP is also superior to ferritin, CRP, and fibribogen in the prediction of sepsis, treatment escalation, and readmission [6].

Two studies are classified as both diagnostic and prognostic. Loots et al. 2022 perform a prospective study in a primary care setting [15]. The primary outcome of this study is the detection of sepsis within 72 h of admission and the secondary prognostic outcome is the ICU admission within 72 h or 30-day mortality. The study of Fisher et al. has already been described above [16].

### 3.3. Settings

The majority of the included studies take place in the ICU. The study of Van Singer et al. takes place in both the ICU and the ED [8]. The study of Loots et al. intends to assess the diagnostic value of biomarkers in septic adults who present to primary care [15]. Three studies by Niggemanet et al., Klein 2020 et al., and Klein 2021 et al. specifically investigate burn patients who suffer from sepsis [20,21,22]. Five studies focus on the prognostic and diagnostic value of PSP in patients who enter the ED. Ultimately, the study of Tschuor et al. is multidivisional, including ED, the department of surgery, and the division of visceral and transplantation surgery [23].

### 3.4. Type of Infections

Two studies involve patients with pneumonia, particularly COVID-19 pneumonia in Lagadinou et al.’s study and Ventilator-Associated Pneumonia (VAP) in Boeck et al.’s study [7,14]. Moreover, the study by Scherr et al. includes patients admitted for AECOPD [12]. Four studies examine the outcomes of sepsis and mortality without highlighting the particular infection sites (Niggeman, Van Singer, Klein 2020, and Fisher 2014 [8,16,20,22]). García de Guadiana-Romualdo 2019 et al. address mostly abdominal infections, whereas abdominal infections exclusively are mentioned in Gukasjan et al.’s 2013 study and Tschuor et al.’s 2012 study [9,11,23]. The study of García de Guadiana-Romualdo et al. in 2018 discusses respiratory and urinary tract infections (UTIs), as well as abdominal infections [18,19]. García de Guadiana-Romualdo 2017 et al. include only respiratory tract infections (RTIs) and UTIs. Michailides et al. refer to IAIs only [6].

Wound infections, central line infections, pneumonia, and UTIs are discussed in Klein 2021 et al. [20]. There is a combination of respiratory, abdominal, UTI, and skin and soft tissue infections (SSTI) in various studies (Hu P, Loots, de Hond, Llewelyn 2013 [5,15,17,26]). Pugin et al. also include bloodstream infections [24]. Infections of the CNS and ENT among pulmonary, abdominal, UTI, bloodstream, and SSTI appear in two studies (Que et al. 2015 and Que et al. 2012) [10,13]. Finally, Klein 2015 et al. examine postoperative infections in patients undergoing cardiac surgery. As a result, they include mediastinitis, peripheral wounds, and sternal wounds among pneumonia and UTIs [25].

## 4. Discussion

Our study interprets the rising significance of PSP as a point-of-care biomarker in both prognostic and diagnostic concepts. The ability of PSP to classify patients with no infection, uncomplicated infection, and sepsis is important in the ED, especially considering the lack of other efficient biomarkers for distinguishing between those situations, as several studies have demonstrated that interleukins, CRP, PCT, and WBC count are not accurate enough to do so [11,18,24]. Its accuracy in bacterial infection diagnosis is proven to be excellent, establishing it as an essential diagnostic tool [27]. Another important factor that renders PSP an essential tool in clinical practice is its sharp increase during a septic episode, leading to early diagnosis. In fact, the increase starts 5 days before a septic episode, which, importantly, is earlier than the increase of other inflammatory biomarkers such as CRP and PCT, reaching peak values within the first 24–48 h [3,20]. The aforementioned characteristics expand the usage of PSP in the risk stratification and prognosis of patients with infection. Emergency departments and ICUs are the most investigated settings for this purpose. Disease severity and clinical course, sepsis development, ICU admission, and mortality are unwanted outcomes that are demonstrated to be predicted early through relying on PSP values [5,10,11,19]. Specifically, in patients with VAP, a low PSP value on day 0 is related to increased survival, whereas a high PSP value on day 7 is related to increased mortality [14]. In patients who undergone cardiothoracic surgery, a 72-h PSP value can predict post-surgery infection and sepsis [25]. In patients with peritonitis, PSP can predict renal failure, MODS, and death [11]. In patients with intra-abdominal infection, PSP can predict readmission, the need for antibiotic treatment escalation, and sepsis development [6].

Despite low quality of evidence as yet, the European Society of Medicine and the Survival Sepsis Campaign propose some cut-off points to use to stratify patients with suspected sepsis based on PSP values for screening and rapid diagnosis. High-risk patients with acute illness and PSP values < 100 ng/dL are at very low risk of complications with septic episodes, whereas values > 300 ng/dL indicate that sepsis is more that 80% likely to occur. In patients with a clinical suspicion of sepsis at the time of assessment, a PSP value < 50 ng/dL can exclude sepsis with >90% NPV [4]. Nevertheless, the heterogeneity in methods and populations of studies used to determine the cut-offs weaken this recommendation. Conversely, PSP measurement combined with other biomarkers, such as CRP, and prognostic scores, such as the SOFA score and the national early warning score (NEWS), is recommended for screening and rapid diagnosis of infection, organ failure, and sepsis [4]. Mai et al. have recently estimated the pooled sensitivity and specificity of PSP in detecting sepsis (88% and 78%, respectively), but there are some fields of heterogeneity such as population and cut-off values that suggest that further research is needed [28]. A recent meta-analysis in an ICU population for the prognostic value of PSP demonstrated that it can predict ICU mortality in adults with infection (AUC 0.69, 95% CI 0.64–0.74) with 90% NPV u the threshold of 133.6 ng/mL. At the threshold of 61.7 ng/mL, PSP discriminated mild from severe infection in terms of sepsis and septic shock (AUC 0.80, 95% CI 0.75–0.85) [29].

There are also some limitations in the establishment of PSP as a point-of-care biomarker in clinical practice. PSP is not tested yet for specific types of infections, such as urinary tract infections (UTI), respiratory tract infections (RTI), and central nervous system infections (CNSI). UTIs, RTIs, and CNSIs are common sources of bacteremia and sepsis, yet there is still no evidence that PSP can predict the clinical course of patients with such diagnoses. There is also a lack of evidence for the prediction of common unfavorable outcomes of infections, such as antibiotic exposure, fever recurrence, and the need for invasive treatment. Furthermore, as it is not constantly measured, there are no data that could connect its kinetics with patients’ clinical course, in contrast with other biomarkers such as CRP and WBC count that are widely used for surveillance of hospitalized patients with infection such that clinicians are experienced and familiar with their fluctuation. Meanwhile, a prospective observational multi-center cohort study is counting daily PSP values to monitor ICU patients at risk of developing sepsis, aiming to determine whether the first-day measurement could predict sepsis development (NCT04105699). Additionally, PSP kinetics in specific populations, such as pregnant women, patients in hemodialysis, immunocompromised patients, and patients with chronic inflammation or patients with previous antibiotic or anti-inflammatory therapy exposure have not been studied yet [4]. In conclusion, there is burgeoning evidence that PSP is a useful point-of-care biomarker for the emergency department due to its ability to recognize bacterial infection and sepsis early, forewarning clinicians about a patient’s clinical course, and for the ICU to recognize the onset of a septic episode early, especially from repeated measurements, which increase diagnostic accuracy. Nevertheless, there is a need to examine PSP’s kinetics and utility in specific populations and conditions to shape the whole picture.

## Figures and Tables

**Figure 1 ijms-25-06046-f001:**
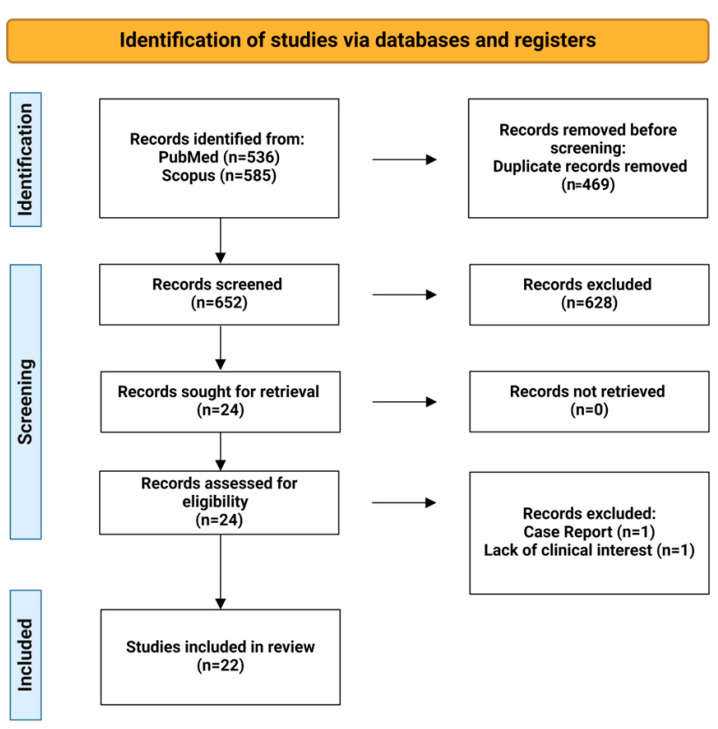
Flow diagram of selection of studies for inclusion.

**Table 1 ijms-25-06046-t001:** Diagnostic studies of pancreatic stone protein.

Title	Author	Year	Study Design	Settings	Type	Population				Outcomes
						Number	Age	Sex (no. (%))	Subgroup	
The critical role of pancreatic stone protein/regenerating protein in sepsis-related multiorgan failure [5]	Hu P	2023	Prospective study	ICU	Prognostic	141	Median 61 (IQR: 50–72)	Male 91 (64)		1. MODS 48–72 h after admission 2. 28-day mortality
The role of the pancreatic stone protein in predicting intra-abdominal infection-related complications: A prospective observational single-center cohort study [6]	Michailides C	2023	Prospective study	Department of Internal Medicine	Prognostic	40	Mean 64.2 ± 22.8		Intra-abdominal infection (IAI)	1. Sepsis 2. Hospital readmission 3. Need of treatment escalation 4. Need of surgery treatment 5. Mortality 6. Length of hospital stay (LOS)
The role of pancreatic stone protein as a prognostic factor for COVID-19 patients [7]	Lagadinou M	2022	Prospective study	Department of Internal Medicine	Prognostic	55	Mean 68.8 ± 14	Male (51.9)	COVID-19 patients	1. In-hospital mortality of patients with COVID-19 2. Non-invasive mechanical ventilation
Pancreatic stone protein for early mortality prediction in COVID-19 patients [8]	Van Singer M	2021	Prospective study	ED and ICU	Prognostic	173	Survival group: median 64.0 (IQR: 52.0–75.0), Dead group: median 81.50 (IQR: 70.3–83.3)	Female survival group, 102; death group, 5	COVID-19 patients	1. 7-day mortality 2. 7-day ICU admission
Prognostic performance of pancreatic stone protein in critically ill patients with sepsis [9]	García de Guadiana-Romualdo L	2019	Single-center, prospective, and observational study	ICU	Prognostic	122	Median 62 (IQR: 52–72)	Male, 68 (55.7)		28-day mortality of PSP in critically ill patients with sepsis
Prognostication of mortality in critically ill patients with severe infections [10]	Que YA	2015	Prospective study	ICU	Prognostic	158 Verification Group, 91 Validation Group	Verification group: mean 61.2 ± 18.2, Validation group: mean 59.9 ± 16.1	Derivation group, female, 93 (65); Validation group, female 54 (37)		Hospital mortality
Pancreatic stone protein predicts outcome in patients with peritonitis in the ICU [11]	Gukasjan R	2013	Prospective study	ICU	Prognostic	91	Median 66 (IQR: 50–72)	Male, 53 (58); Female, 38 (42)		1. Organ failure 2. Multiorgan failure 3. Death in the ICU
Pancreatic stone protein predicts positive sputum bacteriology in exacerbations of COPD [12]	Scherr A	2013	Prospective, monocentric study	ED	Prognostic	200	Median 70 (IQR: 42–91)	Male, 114 (57)	Exacerbations of COPD	Lung bacterial infection in AECOPD
Pancreatic stone protein as an early biomarker predicting mortality in a prospective cohort of patients with sepsis requiring ICU management [13]	Que YA	2012	Prospective cohort study	ICU	Prognostic	107	Mean 59 ± 17.5	Male, 93; Female, 65		Hospital mortality
Pancreatic stone protein: a marker of organ failure and outcome in ventilator-associated pneumonia [14]	Boeck L		Retrospective study	ICU	Prognostic	101	Median 57 (IQR: 43–70)	Male 74	Ventilator-associated pneumonia (VAP)	1. Organ failure 2. Mortality in VAP
Added diagnostic value of biomarkers in patients with suspected sepsis: A prospective cohort study in out-of-hours primary care [15]	Loots FJ	2022	Prospective study	Primary care	Diagnostic and Prognostic	336	Median 80 (IQR: 74–85)	Male, 123 (60); Female; 83 (40)	Septic patients	1. Sepsis (no. = 141) within 72 h of inclusion 2. ICU admission within 72 h or 30-day mortality
Pancreatic stone protein (PSP) and pancreatitis-associated protein (PAP): a protocol of a cohort study on the diagnostic efficacy and prognostic value of PSP and PAP as postoperative markers of septic complications in patients undergoing abdominal surgery (PSP study) [16]	Fisher OM	2014	Prospective monocentric cohort study	Surgical ICU	Diagnostic and Prognostic	160	Unclear	Unclear	Unclear	1. Sepsis 2. Mortality

**Table 2 ijms-25-06046-t002:** Prognostic studies of pancreatic stone protein.

Title	Author	Year	Study Design	Settings	Type	Population				Outcomes
						Number	Age	Sex (no. (%))	Subgroup	
Pancreatic stone protein as a biomarker for sepsis at the emergency department of a large tertiary hospital [17]	de Hond TAP	2022	Semi-prospective, observational cohort study, mono-center	ED	Diagnostic	156	Median 60.0 (IQR: 44.5–73.0)	Male, 82 (52.6)		Sepsis diagnosis (no. = 26)
Incidence and time point of sepsis detection as related to different sepsis definitions in severely burned patients and their accompanying time course of pro-inflammatory biomarkers [20]	Niggemann P	2021	Retrospective study	Burn Center	Diagnostic	90	Mean 48.5 ± 18.8	Female, 18 (20); Male, 72 (80)	Severely Burned Patients	1. Sepsis-3 (no. = 46) 2. Sepsis ABA 2007 (no. = 33) 3. Sepsis Zurich Burn Center (no. = 24)
Serial measurement of pancreatic stone protein for the early detection of sepsis in intensive care unit patients: a prospective multicentric study [24]	Pugin J	2021	Prospective observational clinical study	ICU	Diagnostic	243	Median 65.0 (IQR: 54.0–73.0)	Female, 90 (37); Male, 153 (63)		Sepsis
Response of routine inflammatory biomarkers and novel Pancreatic Stone Protein to inhalation injury and its interference with sepsis detection in severely burned patients [22]	Klein HJ	2020	Longitudinal, observational study	Burn Center	Diagnostic	90	Median 52 (IQR: 9)	Female, 18 (20); Male, 72 (80)	Inhalation Injury no. = 27, ARDS (32%)	Sepsis
Pancreatic Stone Protein Predicts Sepsis in Severely Burned Patients Irrespective of Trauma Severity: A Monocentric Observational Study [21]	Klein HJ	2021	Observational study	Burn Center	Diagnostic	90	Mean 48.5 ± 18.8	Female, 18 (20); Male, n 72 (80)	Severely burned patients	1. Sepsis 2. Infection
Analyzing the capability of PSP, PCT and sCD25 to support the diagnosis of infection in cancer patients with febrile neutropenia [19]	García de Guadiana-Romualdo L	2018	Single-center prospective observational cohort study	ED	Diagnostic	105	Median 63 (IQR: 50–70)	Male, 43 (37.7)	Cancer patients with chemotherapy-associated febrile neutropenia (FN)	Infection
Pancreatic stone protein and soluble CD25 for infection and sepsis in an emergency department [18]	García de Guadiana-Romualdo L	2017	Prospective observational study	ED	Diagnostic	152	Median 66 (IQR: 33)	Male, 88 (57.9)		1. Sepsis 2. Infection
Pancreatic stone protein predicts postoperative infection in cardiac surgery patients irrespective of cardiopulmonary bypass or surgical technique [25]	Klein HJ	2015	Prospective, single-center cohort study	Cardiosurgical ICU	Diagnostic	120	Median 66.5 (IQR: 54.2–75.0)	Female, (27); Male (73)		Infection
Sepsis biomarkers in unselected patients on admission to intensive or high-dependency care [26]	Llewelyn MJ	2013	Observational study	ICU, high-dependency care	Diagnostic	219	Median 65.9 (IQR: 52.0–76)	Female, 93 (42)		1. Sepsis diagnosis 2. Discrimination severe sepsis from non-infective SIRS
The value of pancreatic stone protein in predicting acute appendicitis in patients presenting at the emergency department with abdominal pain [23]	Tschuor C	2012	Prospective, multi-center, cohort study, clinical Trial	ED, Department of Surgery, Division of Visceral and Transplantation Surgery	Diagnostic	245 (Interim analysis will be performed once 123 patients are recruited.)	Unclear	Unclear		Acute appendicitis diagnosis
Added Diagnostic Value of Biomarkers in Patients with Suspected Sepsis: A Prospective Cohort Study in Out-Of-Hours Primary Care [15]	Loots FJ	2022	Prospective study	Primary care	Diagnostic and Prognostic	336	Median 80 (IQR: 74–85)	Male, 123 (60); Female, 83 (40)	Septic patients	1. Sepsis (no. = 141) within 72 h of inclusion 2. ICU admission within 72 h or 30-day mortality
Pancreatic stone protein (PSP) and pancreatitis-associated protein (PAP): a protocol of a cohort study on the diagnostic efficacy and prognostic value of PSP and PAP as postoperative markers of septic complications in patients undergoing abdominal surgery (PSP study) [16]	Fisher OM	2014	Prospective monocentric cohort study	Surgical ICU	Diagnostic and Prognostic	160	Unclear	Unclear	Unclear	1. Sepsis 2. Mortality

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
