# Peer review of "Diagnostic and Prognostic Ability of Pancreatic Stone Protein: A Scoping Review"

_ijms, 2024, doi:10.3390/ijms25116046_

Round 1
Reviewer 1 Report
Comments and Suggestions for Authors
In this review article, the authors tried to systematically review all studies examining a possible connection of pancreatic stone protein (PSP) levels with the severity and possible complications during and after hospitalization of patients diagnosed with infection.
Comments:
The reviewer has some concerns as follows:
1. The novelty of the subject of this review article is not high. Similar review articles have been published recently. For examples, Fidalgo et al. have recently reviewed the clinical evidence available for PSP in the diagnosis and prognosis of sepsis. The selection of studies to describe pancreatic stone protein function, diagnostic and prognostic ability was conducted, and the studies included in review have 23 (n=23) (J Clin Med. 2022 Feb 18;11(4):1085. doi: 10.3390/jcm11041085.); Mai et al. have recently indicated that PSP demonstrates favorable diagnostic accuracy in detecting sepsis, and the studies included in review have 9 (n=9) and studies for pooled analyses have 5 (n=5) (BMC Infect Dis. 2024 May 6;24(1):472. doi: 10.1186/s12879-024-09347-4.). The authors need to clearly state the differences between this manuscript and previous papers, and what are the new findings?
2. The time course or duration for searches and data collection can be shown in the Methods.
3. In this review article, the authors described that it concluded with 22 studies, and included eleven diagnostic studies (Table 1), and included 10 prognostic studies (Table 2). However, there are 12 studies listed in each of the two tables. It needs to be clarified.
4. The references to related and previous works can be strengthened.
5. Overall, this manuscript needs to clarify whether it is a repetitive study before its value can be judged.
Author Response
Reviewer 1
In this review article, the authors tried to systematically review all studies examining a possible
connection of pancreatic stone protein (PSP) levels with the severity and possible complications during and after hospitalization of patients diagnosed with infection.
Comment 1: The novelty of the subject of this review article is not high. Similar review articles have been published recently. For examples, Fidalgo et al. have recently reviewed the clinical evidence available for PSP in the diagnosis and prognosis of sepsis. The selection of studies to describe pancreatic stone protein function, diagnostic and prognostic ability was conducted, and the studies included in review have 23 (n=23) (J Clin Med. 2022 Feb 18;11(4):1085. doi: 10.3390/jcm11041085.); Mai et al. have recently indicated that PSP demonstrates favorable diagnostic accuracy in detecting sepsis, and the studies included in review have 9 (n=9) and studies for pooled analyses have 5 (n=5) (BMC Infect Dis. 2024 May 6;24(1):472. doi: 10.1186/s12879-024-09347-4.). The authors need to clearly state the differences between this manuscript and previous papers, and what are the new findings?
Response to comment 1: This is a scoping review for both diagnostic and prognostic ability of PSP. Thus, we intended to critically approach the current literature, reviewing its size, scope strengthens and limitations in a systematic way. Fidalgo et al.’s narrative review includes some of our references, but it is descriptive and subjective. We added the interesting study of Mai et al. in Discussion section that refers to the diagnostic part. To our knowledge it is the first scoping review on this subject and the first systematic approach that includes both the diagnostic and prognostic aspects of PSP usage.
Comment 2: The time course or duration for searches and data collection can be shown in the Methods.
Response to comment 2: We added the time course of our research in the Methods section.
Comment 3: In this review article, the authors described that it concluded with 22 studies, and included eleven diagnostic studies (Table 1), and included 10 prognostic studies (Table 2). However, there are 12 studies listed in each of the two tables. It needs to be clarified.
Response to comment 3: We clarified that 10 studies are diagnostic; 10 studies are prognostic and 2 are both prognostic and diagnostic.
Comment 4: The references to related and previous works can be strengthened.
Response to comment 4: We added two interesting meta-analyses in the discussion section, one for PSP’s diagnostic ability and one for PSP’s prognostic ability in ICU patients.
Comment 5: Overall, this manuscript needs to clarify whether it is a repetitive study before its value can be judged.
Response to comment 5: This is not a repetitive study but more of a different kind of study which bolsters literature through its scoping character and inclusion of both diagnostic and prognostic studies.
Reviewer 2 Report
Comments and Suggestions for Authors
The authors present a scoping review in which they have examined the existing literature to determine what available evidence is there with regards to correlating the clinical presence of pancreatic stone protein and the severity of pancreas infection. In their searches, the authors found 23 studies in total; 11 evaluating the presence of the protein from a diagnostic perspective, and 12 evaluating such from a prognostic perspective. While those studies identified had variability in terms of the clinical population, forms of infection, and otherwise, the evidence in those 23 articles appears to suggests that pancreatic stone protein does have promise as a biomarker for infection state, with possible application in the bacterial and sepsis context.
In reviewing the manuscript, I made a couple of broad observations. The following should be considered by the authors when preparing a suitable revision.
1. While the writing of the piece is for the most interpretable, there are several grammatical and language errors within the piece that require attention. There are too many to list individually, but the authors should revise the manuscript closely and address these issues before resubmitting.
2. The writing style/structure could also be improved. The authors do employ headings to discuss some aspects of the research identified as pertinent to the study, however, for the main body of text the text is largely descriptive and there could be more discussion of similarities/differences between the pieces. The are headings later on as mentioned, but this could be employed more throughout the article to improve the aspect of analysis of the literature.
Comments on the Quality of English LanguageAs mentioned, there are several aspects of the writing that warrnat attention. These are listed in the main review of the piece in detail.
Author Response
Reviewer 2
The authors present a scoping review in which they have examined the existing literature to determine what available evidence is there with regards to correlating the clinical presence of pancreatic stone protein and the severity of pancreas infection. In their searches, the authors found 23 studies in total; 11 evaluating the presence of the protein from a diagnostic perspective, and 12 evaluating such from a prognostic perspective. While those studies identified had variability in terms of the clinical population, forms of infection, and otherwise, the evidence in those 23 articles appears to suggest that pancreatic stone protein does have promise as a biomarker for infection state, with possible application in the bacterial and sepsis context. In reviewing the manuscript, I made a couple of broad observations. The following should be considered by the authors when preparing a suitable revision.
Comment 1: While the writing of the piece is for the most interpretable, there are several grammatical and language errors within the piece that require attention. There are too many to list individually, but the authors should revise the manuscript closely and address these issues before resubmitting.
Response to comment 1: We carefully revised the manuscript and there were indeed some language issues throughout that were addressed one by one.
Comment 2: The writing style/structure could also be improved. The authors do employ headings to discuss some aspects of the research identified as pertinent to the study, however, for the main body of text the text is largely descriptive and there could be more discussion of similarities/differences between the pieces. The are headings later on as mentioned, but this could be employed more throughout the article to improve the aspect of analysis of the literature.
Response to comment 2: We believe that the type of the structure we present in the paper is more suitable for the reader to identify the meaningful aspect of this work. It would be easier for someone to find the similarities and differences between the pieces as you mentioned based on the used writing style. We believe that study by study analysis is more of technical than descriptive and illustrates the depth of the literature.
Round 2
Reviewer 1 Report
Comments and Suggestions for Authors
This revised manuscript has a great improvement. No further comments.
Reviewer 2 Report
Comments and Suggestions for Authors
The authors have provided adequate repsonses to my initial concerns.